# The Small-Molecule E26-Transformation-Specific Inhibitor TK216 Attenuates the Oncogenic Properties of Pediatric Leukemia

**DOI:** 10.3390/genes14101916

**Published:** 2023-10-08

**Authors:** Ritul Sharma, Chunfen Zhang, Aru Narendran

**Affiliations:** Department of Oncology, Cumming School of Medicine, University of Calgary, 3330 Hospital Dr NW, Calgary, AB T2N 4N1, Canada

**Keywords:** pediatric leukemia, AML, B-ALL, ETS factors, SPI1, PU.1, TK216, venetoclax, 5-Azacitidine

## Abstract

The E26-transformation-specific (ETS) transcription factors regulate multiple aspects of the normal hematopoietic system. There is an increasing body of evidence suggesting aberrant ETS activity and its contribution to leukemia initiation and progression. In this study, we evaluated the small-molecule ETS inhibitor TK216 and demonstrated its anti-tumor activity in pediatric leukemia. We found TK216 induced growth inhibition, cell cycle arrest and apoptosis and inhibited the migratory capability of leukemic cells, without significantly inhibiting the cell viability of normal blood mononuclear cells. Priming the leukemic cells with 5-Azacitidine enhanced the cytotoxic effects of TK216 on pediatric leukemia cells. Importantly, we found purine-rich box1 (PU.1) to be a potential target of TK216 in myeloid and B-lymphoid leukemic cells. In addition, TK216 sharply decreased Mcl-1 protein levels in a dose-dependent manner. Consistent with this, TK216 also potentiated the cytotoxic effects of Bcl-2 inhibition in venetoclax-resistant cells. The sustained survival benefit provided to leukemic cells in the presence of bone-marrow-derived conditioned media is also found to be modulated by TK216. Taken together, our data indicates that TK216 could be a promising targeted therapeutic agent for the treatment of acute myeloid and B-lymphoid leukemia.

## 1. Introduction

Leukemia is the most commonly diagnosed cancer in children. Several factors contribute to the development of pediatric leukemia. These include predisposing genetic conditions, germline and somatic mutations involving translocations and point mutations in critical transcription factors as well as additional cooperating mutations [1]. A single primary genetic alteration or series of such events can occur during any step of hematopoiesis, disrupting normal blood cell development and subsequently leading to leukemogenesis. Pediatric leukemia is a heterogeneous disease, and some of the commonly observed genetic abnormalities that contribute to the development of leukemia in children include KMT2A rearrangements, fusions like ETV6-RUNX1 and BCR-ABL1, hyperdiploidy and mutations in critical genes like *TP53*, *FLT3* and *RAS* [1].

Transcription factors regulate the development of blood cells by controlling lineage determination and through the differentiation of progenitor cells at various stages of hematopoiesis [2]. Dysregulation of transcription factor expression via genetic alterations is known to contribute to leukemogenesis [3]. The ETS family of transcription factors is a large family of proteins that was discovered in the early 1980s. These transcription factors play a crucial role in the normal physiology of the cells by positively and negatively regulating various cellular processes like apoptosis, tissue remodeling and cell cycle regulation [4]. Abnormal activity of ETS factors contributes to the development of oncogenesis, and evidence from several studies has demonstrated their involvement in tumor growth, progression and metastasis [4]. In hematological malignancies, the aberrant activity of the ETS factors is frequently observed to result from abnormal ETS expression and chromosomal translocations leading to oncogenic gene fusions [5,6]. Deregulated expression of multiple ETS factors like *ERG*, *FLI1* and *SPI1* contribute to the genesis of leukemia and impact the survival outcome in leukemia patients [7,8,9].

Targeting transcription factors like the members of the ETS family has been considered challenging as they do not have well-defined active sites for small-molecule inhibitors to bind to. Therefore, the synthesis of ETS inhibitors and their translation into clinical applications has proven difficult. Hence, several novel strategies have been explored to target these transcription factors, including disrupting crucial protein–protein interactions [10]. TK216, a small-molecule ETS inhibitor initially designed to target an oncogenic fusion protein (EWS-FLI1) in Ewing sarcoma, is currently being studied in a phase 2 clinical trial (clinicaltrials.gov, NCT No. NCT05046314). It disrupts the interaction between the ETS fusion protein and RNA helicase, which leads to tumor regression [11]. TK216 is a derivative of YK-4-279, which is known to target EWS-FLI1-harboring Ewing sarcoma and ETV1- and ERG-driven prostate cancer [11,12,13]. The anti-tumor activity of TK216 is also observed in lymphoma, where it exerts its cytotoxic effects by disrupting the interaction between SpiB/Spi1 and RNA helicases, DDX5 and DHX9 [14]. We hypothesized that the therapeutic potential of ETS inhibitors targeting EWS-FLI1 (Ewing sarcoma) and ERG-driven tumors (prostate cancer) can be expanded to pediatric leukemia as they are well known to have deregulated ETS activity.

In the present study, we report the anti-tumor effect of TK216 in pediatric acute myeloid leukemia (AML) and B-acute lymphoid leukemia (B-ALL) cells without significant toxicity in non-malignant peripheral blood mononuclear cells (PBMCs) from healthy individuals. We demonstrate the ability of TK216 to induce cell death and inhibit the migration capacity of these leukemic cells. Importantly, mechanistically, we found purine-rich box1 (PU.1), a blood cell lineage specifying a transcription factor encoded by *SPI1*, to be a potential target of TK216 in pediatric AML and B-ALL cells. Evidence from previous studies has shown PU.1 to be involved in FLT3-mutated as well as KMT2A-rearranged AML and B-ALL leukemogenesis [15,16,17,18]. In addition, inferior overall survival is also observed in AML leukemia patients with high *SPI1* expression [7]. Therefore, we focused our study on targeting PU.1 in pediatric AML and B-ALL cells that carry similar mutations and translocations.

Furthermore, we investigated the potential of combining TK216 with other novel anti-leukemic agents to achieve an enhanced cytotoxic effect on pediatric leukemic cells. Venetoclax, a BH3 mimetic, targets Bcl-2, an anti-apoptotic protein that is commonly overexpressed in AML [19]. However, resistance to venetoclax is observed in AML due to overexpression of *MCL-1* [19]. We show the potential of combining TK216 with Bcl-2 inhibition for enhanced cytotoxic activity in venetoclax-resistant pediatric AML cells. Additionally, we found that priming leukemic cells with a low dose of hypomethylating agent 5-Azacitidine (5-Aza) also enhances the cytotoxic effect of TK216.

Collectively, our data show the pre-clinical evidence of targeting aberrant *SPI1* expressing AML and B-ALL cells with TK216 for future clinical trials. This present report aims to identify the potential of new therapeutics and thus specifically concentrates on the applicability of ETS inhibition for relapsed cell populations.

## 2. Materials and Methods

### 2.1. Cell Lines and Cell Culture

Pediatric leukemia cell lines MV4-11, THP-1 and SUP-B15 were purchased from the American Type Culture Collection (ATCC), Manassas, VA, USA. The KOPN-8 cell line was obtained from the Leibniz Institute German Collection of Microorganisms and Cell Cultures (DSMZ), Germany. The REH cells were a kind gift from Dr. Robichaud’s lab at Université de Moncton (Edmundston, NB, Canada). The PBMCs were isolated from the peripheral blood of pediatric patients diagnosed with leukemia or from the peripheral blood of healthy volunteers, as described previously [20]. All cell lines and primary patient samples were maintained in RPMI-1640 with 10% (*v*/*v*) heat-inactivated fetal bovine serum (FBS) in a humidified incubator at 37 °C with 5% CO_2_. The general characteristics of each cell line and primary patient samples are described in Table 1.

### 2.2. Small-Molecule Inhibitors

The ETS inhibitor TK216, 5-Azacitidine and venetoclax were purchased from MedChemExpress (Monmouth Junction, NJ, USA). The stocks of all the compounds were dissolved in DMSO, and aliquots of 10 mM stock were stored at −20 °C.

### 2.3. Cell Viability Assay

Pediatric leukemia cell lines and patient samples (1 × 10^4^ leukemic cells) were plated in 96-well plates in 100 µL of RPMI-1640 media with 10% FBS. TK216 was diluted in 100 µL of media and added to the cells in varying concentrations (4–0.06 µM). Cells were incubated with this compound for 72 h. All concentrations were tested in triplicate, and cell viability was measured by alamar blue assay (Thermo Fisher Scientific, Waltham, MA, USA). Percent cell viability was calculated from two to three separate biological repeats.

### 2.4. Immunoblotting

Briefly, 1 × 10^6^ leukemic cells were lysed in 100 µL of ice-cold RIPA (radioimmunoprecipitation assay) buffer. The suspension was kept on ice for 10 min and then centrifuged at 15,700× *g* at 4 °C for 15 min. The supernatant was then transferred to pre-chilled tubes, and protein estimation was carried out with an RC DC protein assay by BIO RAD (Hercules, California, USA). The protocol for immunoblotting has been described previously [20]. The following primary antibodies were used in this study: anti-PU.1 (2258S, 1:1000; Cell Signaling, Danvers, MA, USA), anti-Ets-1 (N-276, 1:1000; Santa Cruz, Dallas, TX, USA), anti-PARP (9542S, 1:1000, Cell Signaling), anti-caspase 3 (9662S, 1:1000, Cell Signaling), anti-Mcl-1 (5453S, 1:1000, Cell Signaling), anti-Bcl-2 (4223S, 1:1000, Cell Signaling) and anti-β-actin (MA1-140, 1:5000; Thermo Fisher Scientific). The Western blots presented are representative of 2–3 independent experiments.

### 2.5. Cell Cycle Analysis

Cell cycle analysis was performed using propidium iodide (PI) staining according to established protocol (Abcam, Cambridge, UK). Cells were incubated with DMSO or TK216 (1 µM) for 24 h. Leukemic cells were fixed in 70% ethanol for 30 min at 4 °C. Cells were then washed twice in PBS and treated with ribonuclease. An amount of 200 µL of PI dye was added, and analysis of the results was performed via flow cytometry.

### 2.6. Cell Migration Assay

The effect of ETS inhibition on the migration capacity of leukemic cells was studied using a 6.5 mm diameter Transwell membrane with an 8 µm pore size (Corning, NY, USA). Leukemic cells were treated with TK216 and vehicle control for 24 hours prior to the migration assay. The cells were then washed in PBS. To the upper chamber of the insert, 1 × 10^5^ leukemic cells were added in 150 µL of RPMI-1640 media with no FBS. The lower chamber had 600 µL of RPMI-1640 medium with 10% FBS. The migration assay was performed for five hours, after which the viable migrated cells in the bottom chamber were counted using a trypan blue exclusion assay. Microscopic images were taken under 10× magnification on an EVOS FL auto-imaging system (Thermo Fisher Scientific, Waltham, MA, USA).

### 2.7. Preparation of Bone-Marrow-Derived Conditioned Media (BM-CM)

Bone-marrow-derived stromal cells from a pediatric ALL patient were cultured in RPMI-1640 supplemented with 10% FBS. After two days in culture, the supernatant from the stromal cells was removed and centrifuged at 200× *g* for 5 min and filtered through a 0.45 µM Acrodisc^®^ syringe filter (Cytiva, Marlborough, USA) to remove all contaminating cells and used as a bone-marrow-stroma-derived conditioned medium (BM-CM).

### 2.8. Growth Inhibition of Leukemic Cells Cultured in BM-CM

Leukemia cells (1 × 10^4^ per well) were cultured in a 96-well plate with varying percentages of BM-CM in complete media (RPMI-1640 + 10% FBS) to determine the percentage of conditioned media that leads to maximum leukemic cell viability. Leukemia cells with media alone were used as controls. To investigate the effect of ETS inhibition on BM-CM-stimulated leukemic growth, 1 × 10^4^ leukemia cells in 100 µL of complete media were plated in each well in a 96-well plate. Then, 100 µL aliquots of TK216 in varying concentrations (4–0.06 µM) were added to each well with and without 10% BM-CM. Additional control wells were also set up for leukemia cells in complete media alone and leukemia cells in complete media with 10% BM-CM. Post 72 h incubation, an alamar blue assay was used to assess the differences in cell viability.

### 2.9. Statistical Analysis

Data were plotted as means ± SEM using GraphPad Prism software (Version 10.0.2). Group comparisons were calculated using Student’s *t*-test. For group comparisons, a *p*-value < 0.05 was considered statistically significant.

## 3. Results

### 3.1. Growth Inhibition of Leukemic Cell Lines and Continuously Growing Patient Cells by TK216

Representative pediatric AML and B-ALL cell lines and continuously growing patient samples were used to investigate the cytotoxic activity of TK216 against leukemia cells (Table 1). PBMCs from healthy volunteers were used to measure the baseline toxicity of TK216. In this experiment, leukemic cells were incubated with varying concentrations of TK216 for 72 h, and cell viability was measured with an alamar blue assay. Percent cell viability was calculated after normalization with the DMSO control. As shown in Figure 1a,b, TK216 decreased cell viability in all cell lines and continuously growing primary samples studied. No significant cell death was observed in PBMCs from healthy donors after treatment with TK216 using these concentrations. Moreover, differential sensitivity of cell lines to TK216 was observed with the most sensitive cell line, MV4-11, with an IC_50_ of 0.22 µM, and the least sensitive cell line, SUP-B15, with an IC_50_ of 0.94 µM. The IC_50_ values for each cell line are given in Table 2.

### 3.2. Induction of Apoptosis by TK216 in Pediatric Leukemia Cell Lines

To evaluate the effect of TK216 on cell death, we chose to study the changes in the protein levels of cleaved Poly-ADP ribose polymerase (PARP) and caspase 3 activation, along with anti-apoptotic members of the Bcl-2 family, via immunoblotting. A representative AML cell line, MV4-11, was treated with increasing concentrations of TK216 and the corresponding DMSO control for 24 h. As observed in Figure 2a, an increased amount of cleaved PARP (starting at 0.2 µM) and cleaved caspase 3 (starting at 0.5 µM) was observed with increasing concentrations of TK216, indicating the induction of apoptosis. Levels of the anti-apoptotic protein Mcl-1 were also found to rapidly decrease after treatment. In contrast, the protein levels of Bcl-2 did not change significantly after treatment. This indicates the ability of TK216 to target Mcl-1 in these pediatric leukemia cells. Additionally, cell cycle analysis was performed using propidium iodide staining (Figure 2c). A marked increase in the sub-G_1_ population of cells was observed post treatment, indicating cell death. Moreover, the accumulation of cells in the S phase of the cell cycle was observed after treatment with TK216. Similar experiments were also performed in REH, a representative pediatric B-ALL cell line carrying ETV6-RUNX1 fusion (Figure 2b). Taken together, data from these experiments show that TK216 effectively induced apoptosis and led to a dose-dependent reduction in Mcl-1 protein levels in pediatric AML and B-ALL cells.

### 3.3. TK216 Attenuates the In Vitro Migration Potential of Pediatric Leukemic Cells

After demonstrating the effects of TK216 on leukemic cell viability and cell death, we next wanted to investigate the functional consequence of ETS inhibition on the migration capacity of leukemic cells using a Transwell migration assay. The chemotaxis was induced by the presence of 10% FBS in the lower chamber and leukemic cells (with and without TK216 pre-treatment) were allowed to migrate through the Transwell insert for five hours. Data from these studies show that, as compared to the DMSO control, pre-treatment with TK216 for 24 h effectively reduced the ability of MV4-11 cells to migrate through the membrane of the Transwell insert toward the chemoattractant (Figure 3).

### 3.4. Hypomethylating Agent 5-Azacitidine Sensitizes Pediatric Leukemia Cells to TK216 Treatment

Novel single agents are often used in combination with conventional anti-leukemic agents. Therefore, we evaluated the potential of TK216 to be combined with current anti-leukemic agents. One such class of drugs includes hypomethylating agents such as 5-Aza. Lower doses of hypomethylating agents have been shown to have enhanced anti-tumor effects and clinical response [21]. In this set of experiments, we investigated whether the cytotoxic effect of pre-sensitizing the leukemic cells with a low dose of 5-Aza, followed by TK216 exposure, is superior to that achieved through TK216 treatment alone. A dose–response curve of 5-Aza was produced in pediatric AML and B-ALL cell lines after 72 h of treatment to determine the highest treatment dose of 5-Aza at which leukemic cell viability was not significantly affected (Figure 4a). Based on this, a 50 nM concentration of 5-Aza was chosen to pre-treat the leukemic cells for 24 h. Pediatric leukemic cells with and without prior 24 h treatment with 5-Aza were then exposed to increasing concentrations of TK216 for 72 h. In addition, the effect of 50 nM 5-Aza on cell viability after 24 h was also monitored (Figure 4b). The cell viability was found to decrease more significantly when cells were treated with 50 nM 5-Aza prior to TK216 treatment than TK216 alone (Figure 4c).

### 3.5. PU.1, a Potential Target of TK216 in Pediatric Leukemia Cells

Previously, Spriano and colleagues demonstrated that TK216 disrupts the interaction between Spi1/SpiB and RNA helicases in lymphoma [14]. *SPI1* encodes for PU.1, a protein that plays a central role in normal hematopoiesis as a lineage-specifying transcription factor [22]. Its expression is generally high during myeloid differentiation but decreases during erythrocyte differentiation. Involvement of PU.1 is noted in the leukemogenesis of both B-ALL- and FLT3-mutated and KMT2A-rearranged AML. We hypothesized that PU.1 could be targeted by TK216 in pediatric AML and B-ALL. To test this hypothesis, we used MV4-11, a representative AML cell line carrying FLT3-ITD as well as KMT2A rearrangement, and REH, an ETV6-RUNX1+ B-ALL cell line. Cell lines were treated with multiple doses of the TK216 and DMSO control for 24 h. Our data demonstrate that PU.1 protein levels declined in a dose-dependent manner (Figure 5a–d). In p53-deleted tumors, YK-4-279 has been shown to deregulate the MAPK/ETS-1/p53 signaling axis by targeting ETS-1 [23]. Under the same condition, we evaluated the changes in ETS-1 levels in both MV4-11 and REH cells following TK216 treatment. As compared to PU.1, the protein levels of ETS-1 did not significantly change (Figure 5a,c). Moreover, a negative correlation between the sensitivity of cells to TK216 and PU.1 protein levels was observed in the pediatric AML and B-ALL cells tested (Figure 5e,f). Taken together, our data imply that PU.1 is a potential target for TK216 in pediatric leukemia.

### 3.6. TK216 Potentiates the Cytotoxic Effect of Venetoclax in THP-1 Cells

Bcl-2 has been considered a potential therapeutic target for the treatment of AML [19]. However, it has been previously demonstrated that overexpression of *MCL-1* leads to a lack of response to Bcl-2 inhibitors in AML [19]. Since TK216 decreases Mcl-1 protein levels, we explored the potential of combining TK216 with venetoclax, a Bcl-2 inhibitor in pediatric AML cells. The venetoclax-resistant cell line THP-1 [24] was used for this experiment. Exponentially growing THP-1 cells were treated with varying concentrations of either venetoclax alone or in combination with TK216 (IC_25_ value) for 72 h. Our results show that the addition of TK216 decreased the venetoclax IC_50_ value in THP-1 cells to 1µM compared to the IC_50_ of 8.6 µM obtained with venetoclax alone (Figure 6a). The combination treatment also led to a decrease in Bcl-2 protein levels in THP-1 cells as compared to venetoclax and TK216 treatments alone (Figure 6b,c).

### 3.7. TK216 Overcomes the Leukemia Cell Survival Advantage Provided by BM-Derived Conditioned Media

The bone marrow microenvironment plays a crucial role in leukemia cell survival, where the interaction between the malignant cells and supporting bone-marrow-derived stromal cells provides a survival advantage to leukemic cells and contributes to treatment resistance as well as disease relapse. Presently, however, the details of the mechanisms involved in the promotion of leukemic cells by stromal cells are not well understood. Nevertheless, distinct growth factors secreted by stromata are known to provide support to leukemia cells, which contribute to their persistent survival [25]. Our next set of studies aimed to investigate the potential impact of TK216 with respect to this critical growth regulatory support provided by the bone-marrow-derived stromal cells. Firstly, leukemic cells were cultured in complete media in the presence of various ratios of BM-CM. Our preliminary titration studies identified that 10% BM-CM enhanced the viability of leukemic cells (Figure 7a), and we proceeded with 10% BM-CM for further experiments. Leukemic cells were then treated with varying concentrations of TK216 with and without 10% BM-CM. It was found that, under these experimental conditions, TK216 was able to overcome the survival benefit provided to leukemic cells by bone marrow stromata (Figure 7b). In the presence of 10% BM-CM, TK216 at a concentration of 4µM was able to reduce the leukemic cell viability by 65% in SUP-B15 cells, 78% in KOPN8 cells, 60% in THP-1 cells and 53% in MV4-11 cells when compared to controls under identical conditions.

## 4. Discussion

The ETS family of transcription factors controls multiple aspects of blood development, including lineage switching and the developmental regulation of various blood cell lineages [22]. Consequently, alterations in ETS activity also contribute to leukemia development and progression. The therapeutic targeting of ETS proteins has been hindered by the lack of specific small-molecule inhibitors against these transcription factors. However, other strategies to target transcription factors have been developed. These include the interruption of key protein–protein interactions involved in tumorigenesis. The ETS inhibitor TK216 and its precursor YK-4-279 were originally designed to disrupt the interaction between an ETS fusion protein and RNA helicase in Ewing sarcoma [11]. Since then, YK-4-279 has been shown to have anti-tumor activity across several solid tumors, like Ewing sarcoma, prostate cancer, neuroblastoma, thyroid cancer, melanoma and lymphoma [11,12,13,14,23,26,27,28]. Since it is well established that leukemia has deregulated ETS activity [6,8,15,16,17], we hypothesized that TK216 will be a viable therapeutic candidate for the treatment of children diagnosed with leukemia. The primary objective that we have achieved in this study is the experimental testing of the proof-of-concept, the in vitro anti-tumor activity, and the efficacy of the current ETS inhibitor TK216 for the treatment of pediatric AML and B-ALL.

We first investigated the cytotoxic potential of TK216 in various pediatric AML and B-ALL cell lines and patient samples, covering frequently observed genetic alterations seen in pediatric patients, like KMT2A rearrangements, FLT3 mutations and gene fusions like ETV6-RUNX1 and BCR-ABL1. The marked difference observed in the viability of PBMCs from healthy volunteers and pediatric leukemia cells after TK216 treatment suggests a favorable therapeutic window. The ability of TK216 to induce cell death via apoptosis was studied using markers such as cleaved PARP, caspase 3 and anti-apoptotic members of the Bcl-2 family. Our data demonstrate that with increasing doses of TK216, the amount of full-length PARP and pro-caspase 3 decreased, whereas the protein levels of cleaved PARP and cleaved caspase 3 increased. Inhibition of Bcl-2 is considered an attractive strategy to target AML. However, resistance to Bcl-2 inhibitors is observed, and it is well established that overexpression of *MCL-1* is one of the factors contributing to resistance [19]. One strategy to overcome resistance to venetoclax is to combine small molecules that target Mcl-1 with Bcl-2 inhibitors. Interestingly, under the experimental conditions used, TK216 depleted Mcl-1 but not Bcl-2 protein levels. We hypothesize that in pediatric AML, TK216 is more effective in targeting Mcl-1-dependent leukemic cells. In line with this notion, we combined the Bcl-2 inhibitor venetoclax with TK216 in the venetoclax-resistant cell line THP-1 to investigate the cytotoxic effects of the combination treatment. Interestingly, we found that TK216 was able to augment venetoclax activity in these cells. Moreover, the combination treatment led to a reduction in Bcl-2 protein levels compared to single agents alone. Our results imply that TK216 with venetoclax could be a potentially effective combination strategy for targeting Mcl-1-dependent leukemia cells.

Additionally, cell cycle analysis of TK216-treated cells showed a marked increase in the sub-G_1_ population of cells. Interestingly, different from previous publications on TK216 and YK-4-279, we found an accumulation of cells in the S phase of the cell cycle post TK216 treatment [14,27]. Although leukemia is primarily a disease of bone marrow, leukemic cells often migrate to various tissues and organs like the brain and the testes [29]. The migration capability of leukemic cells was also found to be altered by ETS inhibition in a dose-dependent manner. Furthermore, one of the key goals in pediatric oncology is to determine drug combinations that have a favorable outcome and offer the benefit of reduced toxicity. Hypomethylating agents are used at very low concentrations for the treatment of hematological malignancies. At lower concentrations, they reprogram the epigenome by hypomethylating the promoters of tumor suppressors [21]. We found that priming the leukemic cells with 5-Aza potentiates the cytotoxic effect of TK216.

Multiple previous studies have shown deregulated ETS activity in leukemia. We focused on the ETS factor *SPI1*, which is a highly expressed transcription factor present during the early stages of hematopoiesis and myeloid differentiation [22]. Previous reports have shown the involvement of PU.1 in KMT2A-rearranged and FLT3-mutated AML as well as B-ALL pathogenesis [15,17,18], and targeting PU.1 has been shown to be an effective therapeutic approach for AML treatment [30]. Spriano and colleagues previously demonstrated that TK216 disrupted SPI1/SPIB interaction with RNA helicases in lymphoma models [14]. We investigated PU.1 as a potential target of TK216 in pediatric AML and B-ALL cells. We observed a marked decrease in PU.1 levels after 24 h of dose-dependent treatment with TK216 in pediatric AML- and B-ALL-representative cell lines. Other ETS factors, like ETS-1, have also been shown to be the target of YK-4-279 [23]. Therefore, under the same conditions, we also investigated the effect of TK216 on ETS-1 and found no significant difference in ETS-1 protein levels after TK216 treatment in MV4-11 and REH cells. Moreover, correlation analysis revealed a negative correlation (r = −0.98) between the amount of PU.1 protein expressed in the pediatric leukemic cells and the IC_50_ value for TK216 in these cells. This further indicates that PU.1 is a potential target of TK216 in pediatric leukemia cells.

Leukemic blast cells originate and exist in the bone marrow along with other normal hematopoietic cells. The interactions between leukemic blast cells and bone marrow stromal cells are quite complex. The stromal cells interact with leukemic cells primarily through secreting soluble factors and physical contact [25]. The stromal cells in the bone marrow provide a growth advantage to leukemic cells and are thought to contribute to relapse. We tested the effect of TK216 on the growth inhibition of leukemic cells in the presence of soluble factors secreted by bone marrow stromal cells. Our data demonstrate that ETS inhibition nevertheless reduced the survival of leukemic cells cultured in 10% BM-CM. Additional studies on the role of PU.1 inhibition in the survival of leukemic cells when co-cultured with stromal cells could further validate these findings.

In conclusion, our data demonstrate the ability of TK216 to induce apoptosis via caspase 3 cleavage, cell cycle arrest and inhibition of migration potential in pediatric AML and B-ALL cells. Importantly, we have shown that the combination treatment of TK216 with venetoclax could overcome Bcl-2 resistance in venetoclax-resistant cells. Furthermore, the finding that TK216 overcomes the survival advantage provided to leukemic cells by bone-marrow-derived conditioned media has important therapeutic significance. Taken together, the findings from our study suggest TK216 is a promising therapeutic agent in pediatric AML and B-ALL and justify further evaluation of this family of inhibitors in future clinical studies for this population of patients.

## Figures and Tables

**Figure 1 genes-14-01916-f001:**
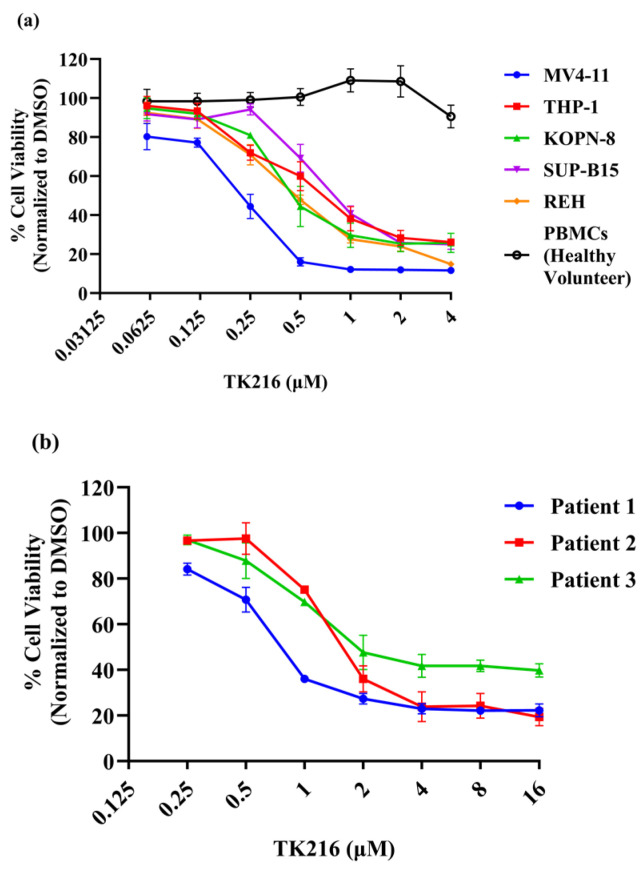
Growth inhibition of pediatric leukemic cells by TK216. (**a**,**b**) TK216 decreased the cell viability in a panel of pediatric leukemia cell lines and continuously growing patient samples in a dose-dependent manner. The cytotoxic effect of TK216 on cell viability of PBMCS from healthy individuals was minimal. Percent cell viability normalized to DMSO was measured via alamar blue assay. Data represented as means ± SEM from 2–3 independent biological experiments.

**Figure 2 genes-14-01916-f002:**
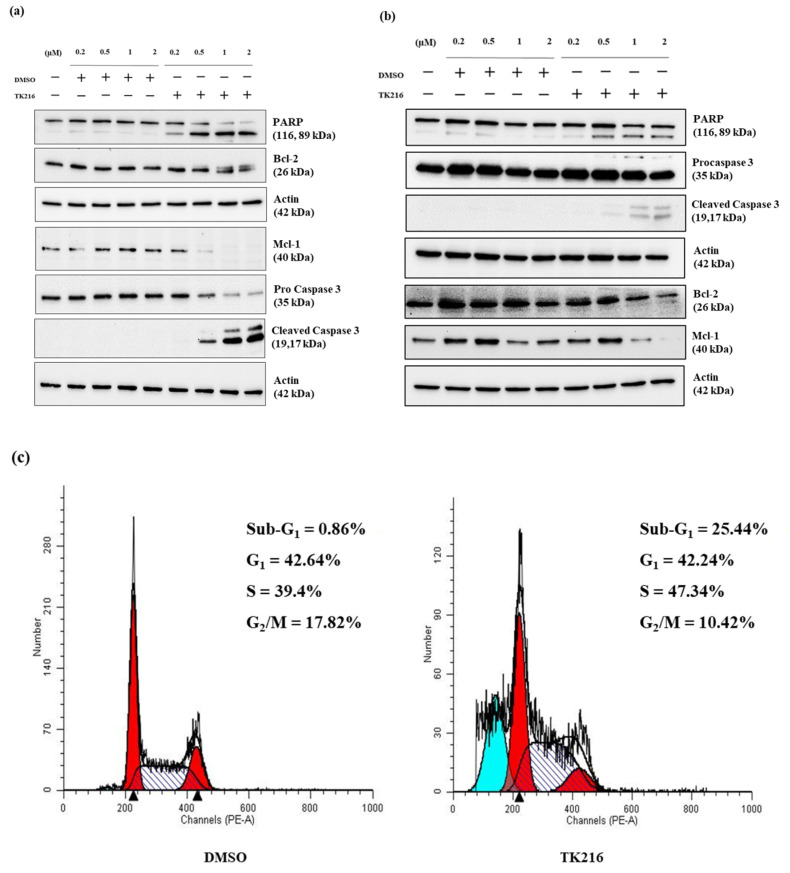
ETS inhibition induces cell death in leukemia cells. (**a**) Apoptosis induction in TK216-treated cell lysates. MV4-11 cell lysates were treated with varying doses of TK216 and corresponding DMSO control for 24 h and analyzed via Western blotting. TK216 led to increased levels of cleaved PARP and cleaved caspase 3 in a dose-dependent manner. Mcl-1 levels were also found to be depleted significantly post treatment. Actin was used as a loading control. (**b**) Representative blot for REH cells treated with TK216 for 24 h. Both MV4-11 and REH blots are representative of 2–3 separate biological experiments. (**c**) Cell cycle analysis via PI staining in TK216-treated leukemia cells. MV4-11 cells were treated with TK216 (1 µM) and DMSO control for 24 h. The different peaks correspond to the percentages of cells in various stages (G_1_, S and G_2_/M) of the cell cycle. The blue peak corresponds to the sub-G_1_ population of cells in treated cell lines.

**Figure 3 genes-14-01916-f003:**
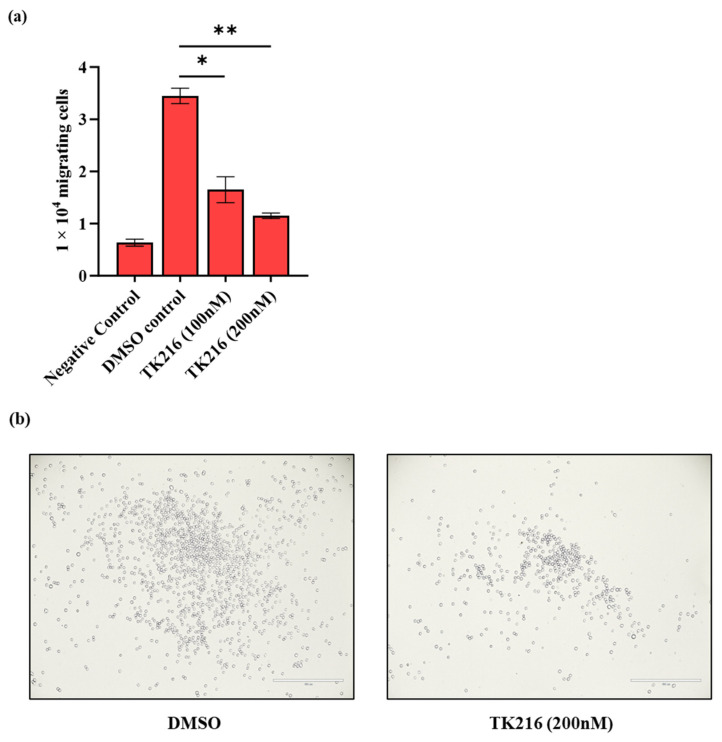
Reduction in migration ability of leukemic cells by TK216 (**a**) MV4-11 cells were treated with TK216 (100 nM, 200 nM) and DMSO control for 24 h. Transwell migration assay was performed for five hours, following which cells were harvested and counted in the bottom chamber. Data represented as mean ± SEM of two technical replicates. Similar results were obtained from two independent biological experiments. The differences in the groups were calculated via Student’s *t*-test. (* *p*  < 0.05, ** *p*  < 0.01). (**b**) Microscopic images taken under 10× magnification. Scale—400 µm.

**Figure 4 genes-14-01916-f004:**
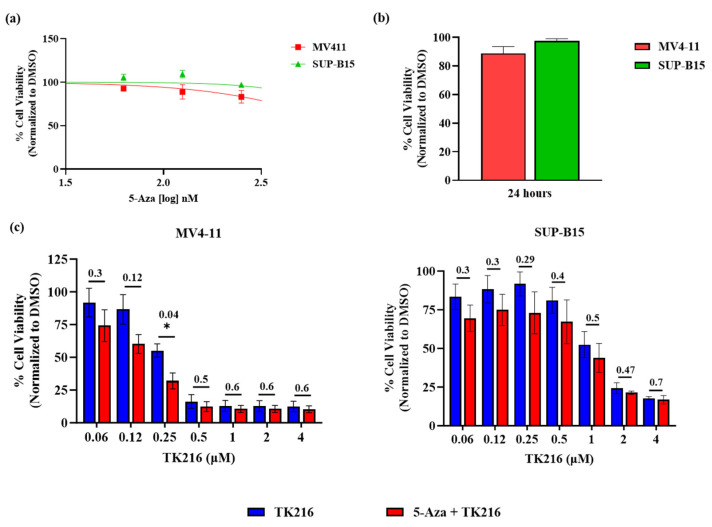
Pre-treatment with 5-Azacytadine sensitizes the leukemic cell lines to TK216 treatment. (**a**) The dose–response curve of 5-Aza in pediatric leukemia cell lines. (**b**) Cell viability post 24 h treatment with 50 nM 5-Aza. (**c**) Pediatric leukemia cell lines treated with TK216 alone and post 24 h treatment with a low dose of 5-Aza. Data represented as mean ± SEM of three independent biological experiments (* *p* <  0.05).

**Figure 5 genes-14-01916-f005:**
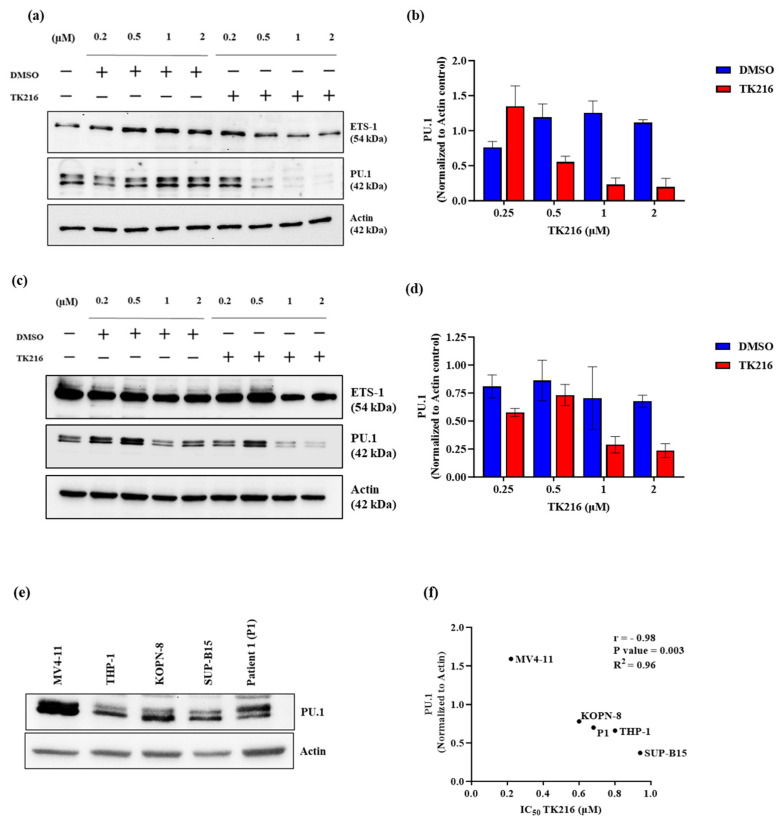
PU.1, a potential target of TK216 in pediatric leukemia cells. (**a**) MV4-11 and (**c**) REH cell lysates treated with varying doses of TK216 and corresponding DMSO control for 24 h and analyzed via Western blotting. Actin was used as a loading control. Blots are representative of 2–3 separate experiments. (**b**,**d**) Densitometry analysis of PU.1 protein levels in MV4-11 and REH cells treated with TK216. (**e**) Protein levels of PU.1 in pediatric leukemia cell lines and patient sample as determined with immunoblotting. (**f**) Correlation between the amount of PU.1 levels expressed by pediatric leukemia cells and their sensitivity to TK216.

**Figure 6 genes-14-01916-f006:**
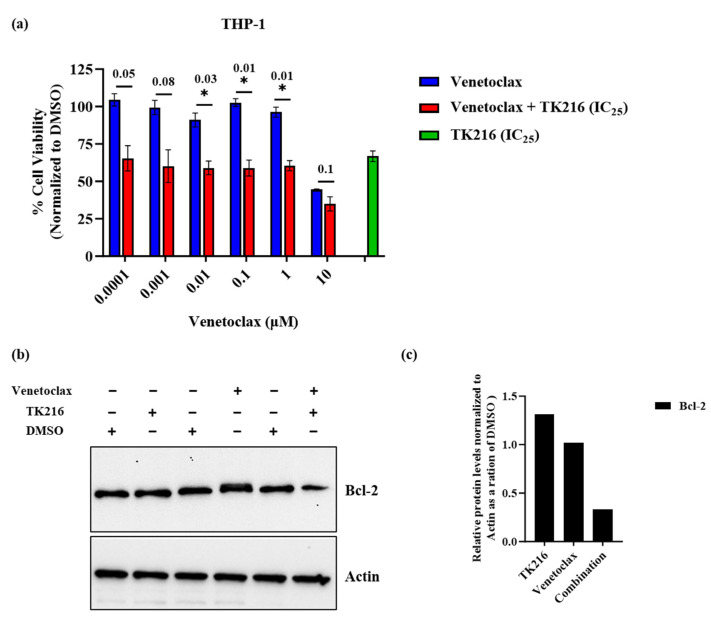
TK216 potentiates the cytotoxic effect of venetoclax in venetoclax-resistant cell line. (**a**) THP-1 cells were treated with multiple doses of venetoclax either alone or in combination with TK216 (IC_25_) for 72 h. Data represented as standard error of the means from two independent biological experiments (* *p* < 0.05). (**b**) The protein levels of Bcl-2 when THP-1 cells were treated with TK216 (IC_50_ value), venetoclax (IC_50_ value) or a combination of both the agents for 24 h. (**c**) Densitometry analysis of Bcl-2 in THP-1 cells treated with TK216, venetoclax or a combination of both the agents.

**Figure 7 genes-14-01916-f007:**
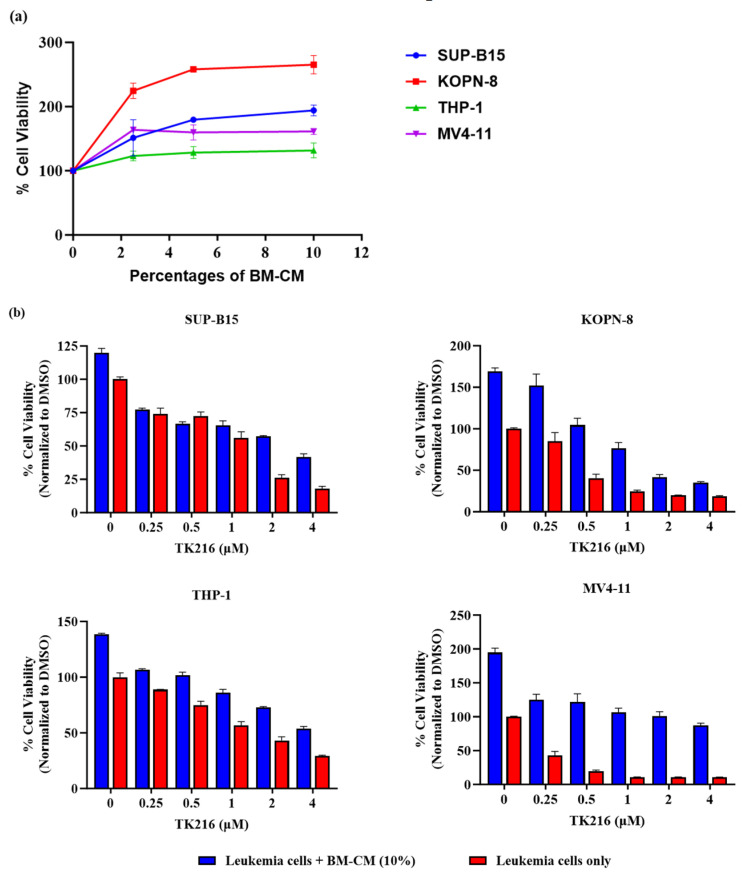
TK216 overcomes the survival advantage provided to leukemic cells by bone-marrow-derived conditioned media. (**a**) The cell viability of leukemic cell lines post 72 h incubation with different percentages of BM-CM. Data represented as means ± SEM from two independent biological experiments. (**b**) Pediatric leukemic cells were cultured with or without 10% BM-CM and treated with TK216 for 72 h. Data represented as means ± SEM from three technical repeats. A similar trend was observed in 2–3 independent biological repeats.

**Table 1 genes-14-01916-t001:** Panel of pediatric leukemia cell lines and continuously growing patient samples used in the study.

Cell Line	Age/Sex	Leukemia Subtype	Genetic Fusions	Additional Genetic Alterations
MV4-11	10Y/M	AML	KMT2A-AFF1	FLT3-ITD
THP-1	1Y/M	AML	KMT2A-MLLT3	NRAS (p.Gly12Asp), TP53 (174-182del)
KOPN-8	3M/F	B-ALL	KMT2A-MLLT1	KRAS (p.Gly12Asp), TP53 (p.Arg248Gln)
SUPB-15	9Y/M	B-ALL	BCR-ABL1	-
REH	15Y/F	B-ALL	ETV6-RUNX1	PTEN (p.Arg173Cys) TP53 (p.Arg181Cys)
Patient 1	8Y/F	AML	-	Hyperdiploidy
Patient 2	3M/M	B-ALL	KMT2A-MLLT1	-
Patient 3	16Y/M	B-ALL	-	-

**Table 2 genes-14-01916-t002:** IC_50_ values for pediatric leukemia cell lines and patient samples after treatment with TK216 for 72 h.

Cell Line	TK216 IC_50_ (µM)
MV4-11	0.22
THP-1	0.79
KOPN-8	0.61
SUPB-15	0.94
REH	0.53
Patient 1	0.68
Patient 2	1.28
Patient 3	1.90

## Data Availability

The data presented in this study are available in The Small-Molecule E26-Transformation-Specific Inhibitor TK216 Attenuates the Oncogenic Properties of Pediatric Leukemia.

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
