# Peer review of "The Small-Molecule E26-Transformation-Specific Inhibitor TK216 Attenuates the Oncogenic Properties of Pediatric Leukemia"

_genes, 2023, doi:10.3390/genes14101916_

Round 1

Reviewer 1 Report

In this article, Sharma et al investigate in vitro anti-tumor activity of TK216 in  pediatric leukemia alone or in combination with other drugs.  They demonstrate on leukemia cell lines and patients’ sample how TK216 are able to induce cell apoptosis and reduce cell migration. The manuscript is well written but some considerations have to be done on the overall design of the study. I suggest the authors review the manuscript according to the following considerations:

-       The authors studied myeloid and lymphoid cell lines and patient samples to cover, as declared in the discussion, the spectrum of leukemia seen in children. However, the low number of cell lines studied cannot be considered representative of the genetic variability of pediatric leukemia. In addition, three out of four cell lines and one patient’s sample carry KMT2A rearrangements, and some experiments (such as those presented in figure 2) have been performed only on t(4;11) leukemia subtype. Thus, to consider the current study representative of pediatric leukemia,  several cell lines  and patients samples carrying other genetic abnormalities should be studied.

I suggest adding some experiments on samples/cell lines carrying other representative fusion transcripts especially ETV6/RUNX1. The authors should consider that E/R positive ALL is the most common leukemia in children, t(12;21) directly involves a gene of the ETS family and RUNX1-PU.1 axis has an important role in normal hematopoiesis and in lymphoid and myeloid leukemia. Furthermore, some data have been published on REH cell lines treated with TK216 by Mehtonen J et al. in 2020  (Genome Med. 2020 Nov 20;12(1):99. doi: 10.1186/s13073-020-00799-2)

-       The authors selected only samples of relapsed patients; have they also considered the study of samples at the diagnosis?

According to my opinion, these major remarks could strengthen the results of the study.

Furthermore, I would have the following minor remarks:

-       Introduction:  A word is probably missing in the following sentence( see highlighted words) : In the present study, we report the anti-tumor effect of TK216 in pediatric leukemia cells without non-specific toxicity non-malignant peripheral blood mononuclear cells (PBMCs) from healthy individuals. 

-       Table 1: For patient 3 in age/sex section KCCF 10 is reported. Can the authors clarify the meaning or report this abbreviation in the legend below?

-       In the manuscript the authors used the abbreviated mins to indicate minutes, but the use of the extended form may be desirable. 

-       Legend Figure 1:  do n° 3 and n° 2 refer to the number of cell lines/samples or to the biological repeats? Please specify. In the first case the data are not consistent with the graph.

-       In most of the graphs the authors do not report data on statistical analysis?  Even when not statistically significant the results of the comparisons  and the p-values should be reported in  the results and fully argued in the discussion.

-       Figure 4: the authors should report statistical analysis on comparison monotherapy vs combination treatment; pre-treatment with 5-aza seems to enhance TK215 activity but statistics should be reported and commented ( see previous consideration).

-       Figure 6b: the graph is not clear in the upper part the + and – symbols are overlapped. Please edit.

-       In section 3.7 the authors explain that their  preliminary titration studies identified that 10% BM-CM enhanced the viability of leukemic cells (Data not shown)….. The authors should report at least one reference about data not shown.

-       All the genes must be written in italics.

-       Please accurately check the references:

1.     The number of each reference is reported twice; please edit.

2.     The references are not reported uniformly, e.g., all authors were reported for reference n° 5 while three authors for the others.

3.     In references n 25, 26 and 27 some authors are wrongly reported or only the first author has been reported in addition the symbol "should be removed.

Reviewer 2 Report

This is very interesting article concerning the small molecule ETS inhibitor TK216 and its potential role in the treatment of pediatric leukemia.

Authors found that TK216 induced growth inhibition, cell cycle arrest, apoptosis and inhibited the migration capability of leukemic cells without significantly inhibiting the cell viability of normal blood mononuclear cells. They revealed that 5-Azacitidine enhanced the cytotoxic effects of TK216 on pediatric leukemia cells and that TK216 potentiated the cytotoxic effects of BCL-2 inhibition in venetoclax resistant cells.

The article is well written. The introduction provides sufficient background, the methods are described clearly. The authors present results in very intelligible way. The discussion is extensive enough. Conclusions are supported by the results. 

Author Response

We would like to thank you for the consideration of our submitted manuscript for publication in Genes. We very much appreciate the careful evaluation of our manuscript by reviewer 2.

No revisions/edits were asked by reviewer 2.

Reviewer 3 Report

Dear Authors,

Leukemias are a heterogeneous group of diseases and genetic changes occurring in individual subtypes have a significant impact on prognosis and response to treatment. You do not take this into account and apply the obtained results to all leukemias in children, which in my opinion is a mistake. The introduction and discussion should be improved in this regard. The introduction lacks information on childhood leukemias to justify the selection of cell lines and patients tested. In the introduction there is also no information about the drugs studied (5-AZA, venetoclacx).

material and methods section

There is no information on the number of repetitions for each experiment. If done singly, the results are unreliable. There is also no information on how many healthy volunteers there were.

In material and methods section is a sentence: “Then, a 100 µl aliquots of TK216 in varying concentrations were added to each well with and without 10% BM. - but concentrations are not specified!

Table 1, patient 3 – KCCF 10 – what is this?

PU.1 - not given the full name on first use or other information explaining what it is.

Discussion - there are stylistic errors in the discussion (e.g. see the first sentence).

Round 2

Reviewer 1 Report

The second version of the manuscript is, in my opinion , significantly improved; the authors have reported additional experiments and studied  other cell lines to increase the strength of the dataAuthors' answers to previous major revisioncan be considered acceptable and all minor revisionhave been resolved. Based on this consideration the manuscript, in this version, can be considered acceptable for publication, however, 2 additional minor issues should be taken into account: 

- The IC50 of patient 3 is, as reported in table 2 equal to 0.98 but this value seems not to be consistent with the graph in figure 1the IC50 for patient appears to be around 2 mM. Can the authors verify?

- In the figure 7 the authors reported the -values of  the comparison between cells colturewith or without CM-BM. These comparisondemonstrate that, as expected, cells cultured with CM-BM, have higher viability than leukaemic cell alone in all treatment groups. I advise the authorto remove all the p-values related to these comparisons because they are not the focus of the experiments; the result to underline is the reduction of leukaemic cell viability, also in BM-CM cell cultures,  with increasing dose of TK216. In my opinion, if the authors wanted to report p-values they would have to compare cell grown in the same conditions  and exposed to different TK216 doses.

Author Response

We would like to thank Reviewer 1 for their input. 

Comment 1: The IC50 of patient 3 is, as reported in Table 2 equal to 0.98 but this value seems not to be consistent with the graph in Figure 1the IC50 for patient 3 appears to be around 2 mM. Can the authors verify this?

Response: Thank you for pointing this out. We have corrected it now. The IC50 value is 1.9µM. (See Table 2 on page 5).

Comment 2: In Figure 7 the authors reported the p-values of the comparison between cells cultured with or without CM-BM. These comparisons demonstrate that, as expected, cells cultured with CM-BM, have higher viability than leukemic cells alone in all treatment groups. I advise the authors to remove all the p-values related to these comparisons because they are not the focus of the experiments; the result to underline is the reduction of leukemic cell viability, also in BM-CM cell cultures,  with increasing dose of TK216. In my opinion, if the authors wanted to report p-values they would have to compare cells grown in the same conditions and exposed to different TK216 doses.

Response 2: We agree with this comment. We have removed all the p-values from Figure 7 (Please see page 11).

Reviewer 3 Report

The authors made corrections in accordance with the review, the article can be published in this form

Author Response

We would like to thank reviewer 3 for their input. 

No further edits/corrections are needed.